# Hydrophilic and Hydrophobic Effects on the Structure and Themodynamic Properties of Confined Water: Water in Solutions

**DOI:** 10.3390/ijms22147547

**Published:** 2021-07-14

**Authors:** Francesco Mallamace, Domenico Mallamace, Sow-Hsin Chen, Paola Lanzafame, Georgia Papanikolaou

**Affiliations:** 1Department of Nuclear Science and Engineering, Massachusetts Institute of Technology, Cambridge, MA 02139, USA; sowhsin@mit.edu; 2Istituto dei Sistemi Complessi, Consiglio Nazionale delle Ricerche, 00185 Rome, Italy; 3Departments of ChiBioFarAm and MIFT, Section of Industrial Chemistry, University of Messina, CASPE-INSTM, V.le F. Stagno d’Alcontres 31, 98166 Messina, Italy; mallamaced@unime.it; 4Departments of ChiBioFarAm, Section of Industrial Chemistry, University of Messina, CASPE-INSTM, V.le F. Stagno d’Alcontres 31, 98166 Messina, Italy; paola.lanzafame@unime.it (P.L.); georgia.papanikolaou@unime.it (G.P.)

**Keywords:** water, local order, relaxation times, self-diffusion, hydrophobic effect

## Abstract

NMR spectroscopy is used in the temperature range 180–350 K to study the local order and transport properties of pure liquid water (bulk and confined) and its solutions with glycerol and methanol at different molar fractions. We focused our interest on the hydrophobic effects (HE), i.e., the competition between hydrophilic and hydrophobic interactions. Nowadays, compared to hydrophilicity, little is known about hydrophobicity. Therefore, the main purpose of this study is to gain new information about hydrophobicity. As the liquid water properties are dominated by polymorphism (two coexisting liquid phases of high and low density) due to hydrogen bond interactions (HB), creating (especially in the supercooled regime) the tetrahedral networking, we focused our interest to the HE of these structures. We measured the relaxation times ($T_{1}$ and $T_{2}$) and the self-diffusion (D${}_{S}$). From these times, we took advantage of the NMR property to follow the behaviors of each molecular component (the hydrophilic and hydrophobic groups) separately. In contrast, D${}_{S}$ is studied in terms of the Adam–Gibbs model by obtaining the configurational entropy (S${}_{conf}$) and the specific heat contributions (C${}_{P,conf}$). We find that, for the HE, all of the studied quantities behave differently. For water–glycerol, the HB interaction is dominant for all conditions; water–methanol, two different T-regions above and below 265 K are observable, dominated by hydrophobicity and hydrophilicity, respectively. Below this temperature, where the LDL phase and the HB network develops and grows, with the times and C${}_{P,conf}$ change behaviors leading to maxima and minima. Above it, the HB becomes weak and less stable, the HDL dominates, and hydrophobicity determines the solution.

## 1. Introduction

Water is one of the most interesting materials to study because of its central role in many research fields with its unusual thermodynamics compared to normal liquids [1]. It is well known to researchers that almost all of its properties have an anomalous behavior as a function of thermodynamic variables, especially in the metastable supercooled liquid regime below its melting temperature $T_{m}$ up to the homogeneous nucleation temperature ($T_{h}$) [2]. The best known of these is the density maximum (ρ) at 277 K, accompanied by unusual behaviors in the pressure (*P*) and temperature (*T*) behaviors of the thermodynamic response functions: in particular, the isobaric specific heat ($C_{P}$), the compressibility (isothermal $\kappa_{T}$ and adiabatic $\kappa_{S}$), and the expansion coefficient ($\alpha_{P}$), all describing local fluctuations in volume (δV) or entropy (δS).

In a regular liquid, these fluctuations are positively correlated and decrease as *T* decreases, whereas for water, below $T_{m}$, they not only grow but also become anti-correlated (an increase in *V* brings an *S* decrease). These behaviors in the supercooled regime are due to a growing development in its local order that, as proposed by Speedy and Angell [3], can create diverging (critical-like) behaviors in the mentioned response functions. For ambient pressure, the diverging temperature was observed for $\kappa_{T}$ at $T_{S}∼228$ K [3].

Another characteristic of water in the solid crystalline phase is polymorphism, i.e., ice has many different structural forms ranging from the ice Ic to ice XIX [4]. After the Mishima discovery of “polyamorphism” [5,6,7], i.e., the existence of glassy forms with different densities, the idea of a liquid polymorphism [8,9,10,11,12] has been confirmed. Specifically, Mishima discovered the water high-density amorphous phase (HDA) using the pressure-amorphization of ice ($I_{h}$), whereas the corresponding low-density (LDA) obtained by the deposition of water vapor onto a very cold substrate was known since 1935 [13]. These two amorphous phases can be transformed into each other using a reversible first order transition [14]. Furthermore, at ambient pressure, LDA if heated, undergoes a glass to liquid transition (at about 130 K) into a highly viscous fluid and then crystallizes at $T_{x}$ = 150 K. Another amorphous water, the VHDA (very high density amorphous), was recently discovered [15].

An extension to the liquid phase of this singular characteristic of amorphous water has improved our knowledge of its properties. Similar to that of glass, liquid polymorphism is due to two phases of different density, i.e., the high- and low-density liquids (HDL and LDL, respectively). The LDL has an “open” structure governed by a networking process with a tetrahedral symmetry due to the noncovalent attractive hydrogen bonding (HB) interaction. Under precise thermodynamic conditions, HDL and LDL can coexist, and by changing pressure or temperature, they can change one into the other by means of a first-order transition: the liquid–liquid transition hypothesis (LLT). An original idea originating from an MD study taking into account the discontinuity of the LDA–HDA transition has become central in water studies as it is the basis of the liquid–liquid critical hypothesis (LLCP or second critical point in distinction to the vapor–liquid one) [16].

In contrast to the HB, in water, there is also an intermolecular Coulomb repulsion between electron lone-pairs on adjacent oxygen atoms and two H–O covalent bonds created by sharing the electron lone pairs. Hence, the HB dominates water in the stable and supercooled regime, and the repulsive lone pairs mainly influence the physics of water from above the boiling temperature ($T_{b}$) in the sub-critical and critical regions. The vapor–liquid critical point CP is located at $T_{C}=647.1$ K at $P_{C}=22.064$ MPa, and the LLCP is estimated by MD experiments to be located near 200 K and at a pressure less than 200 MPa [16,17,18].

For decreasing entropy, the divergent behavior observed in water response functions as well as its liquid polymorphism can explain its anomalies and complexity; through them, the presence of a water molecular tetrahedral local order has been experimentally demonstrated. Unfortunately, in spite of the very large number of accurate computational studies, with their fundamental results [17], the corresponding criticality (inside the supercooled regime) is far from experimentally proven [18]. Today, it seems to be a fascinating chimera for experimental physics, although we are sure of the liquid polymorphism, in particular, of the LDL phase favored by a temperature decrease and the corresponding growth of the hydrophilic interaction represented by the hydrogen bond. As proposed by many simulation studies and experimental data (also developed in confined water), the LDL tetrahedral symmetry is that of ordinary ice, in which each water molecule has four nearest neighbors and acts as a H-donor to two of them and as a H-acceptor for the other two. A *T* decrease involves both a growth in size of its structural networking and its greater stability: the HB lifetime strongly increases (by many orders of magnitude) from picosecond values characteristic of the stable liquid water [19]. However, it must be stressed that, whereas the ice tetrahedral network is permanent, the liquid water tetrahedrality is instead local and transient. It should be noted that a pressure increase contrasts these ordering effects.

In principle, the bulk liquid water cannot exist stably in the region between the homogeneous nucleation temperature ($T_{h}$) and that in which the ultra-viscous liquid obtained from the fusion of LDA crystallizes ($T_{x}$). However, if water is confined in nano-pores smaller than the nucleation centers, this constraint can be overcome and water can be easily maintained in the liquid state in the range *T*${}_{h}$–*T*${}_{x}$ and the LDA can be also achieved [20]. Other ways to explore these low-temperature regions is to study water in solutions; inside ice; or on the outside, as water from the hydration of macromolecules (many of biological interest) and micellar systems [21], or by melting a multi-molecular ice surface [22]. In such a way, many important water properties due to polymorphism were discovered, e.g., the existence of a density minimum [23,24], as predicted by Percy W. Bridgman [25] more than a century ago and subsequently confirmed by computational studies [26,27,28,29,30]. Other important results concern the dynamics of the system such as the crossover from a fragile to a strong glass-forming material, originally predicted by Angell [31] and observable at ambient pressure at $T_{L}≃225$ K [32]; this is also in the water P−T phase diagram and in the locus of the Stokes–Einstein relation violation (due to the onset of dynamic heterogeneities and decoupling between the translational and rotational modes) and of the Widom line. This last line strongly linked to the LLCP (hence to the LDL and HDL) identifies the maximum in the δV and δS fluctuations, where thermodynamic response functions reach their extremes (minimum with negative values in the $\alpha_{P}$ and maxima in $C_{P}\;$and $\kappa_{T}$ [17]).

As known, confined water is involved in a very large part of material systems, in particular, in those of biological interest [33]. In these situations, the hydrogen bond (and the system polymorphism) plays an important role although the hydrophobic interaction is equally fundamental. Hydrophobicity is shown in aqueous solutions by nonpolar substance aggregation, which excludes water, and therefore moieties with these properties characterize amphiphilic molecules. Both of these interactions are of fundamental importance in many fields of science and technology. A relevant example is the role played by both in the folding and unfolding of proteins. Unfortunately, opposite to the well-described hydrophilicity, little is known about hydrophobicity. Amphiphiles are usually organic compounds with a head (polar if ionic or HB if non-ionic) and an apolar aliphatic chain (hydrophobic groups) that makes contact with water molecules that strongly avoid each other [34].

The amphiphilic properties are defined by these two opposite conditions; in water or oil solutions, a single molecule cannot satisfy both while a cluster of molecules can, and building blocks of mesoscopic structures are formed under stable thermodynamic conditions [35]. Many polymers and polyelectrolytes containing both a water-insoluble (or oil-soluble) component and a water-soluble component belong to this class of materials. This is the “soft condensed matter” made of complex mesoscopic materials (such as long helical rods (e.g., polypeptides, DNA, RNA, and proteins) and discoid organic molecules, polymers, colloids, and many different multimolecular-associated structures (membranes and bilayers)) that, despite their complexity, can be described in terms of current statistical physics by means of scaling laws and the concept of universality [36,37].

Past studies, many theoretical and computational (see, e.g., References [38,39,40]) addressed the solute effect on the solvent (structure and energetics [41]), but despite many attempts, we do not yet have any analytical forms for quantitatively treating hydrophobicity. An experimental measurement of the pair distribution function between hydrophobic molecules as well as the corresponding potential of mean force between these two molecules are lacking. Hence, we are unable to understand the forces underlying hydrophobic interactions and to evaluate their implications [42]. New experiments are thus necessary in order to give the basis for a quantitative theory of hydrophobic effect (HE) that enables the study of complex materials including bio-systems. However, just by using water confined in hydrophobic nanotubes, it has been experimentally demonstrated (by means of nuclear magnetic resonance (NMR)) that the hydrophobicity becomes effective only in the high *T* regime (T>281 K) [43].

There are many solutes with chemical moieties that affect the water HB ordering process, e.g., the ion charges in salt solutions or the hydrophobic heads in simple alcohols and polymer systems. At the same time, the macromolecular functions (peptides, proteins, and DNA) are affected by their interaction with water, in particular, by the hydrophilic/hydrophobic contrast, meaning that water is not only a solvent but also an integral and active component. It is itself a sort of “biomolecule” that plays both a dynamic and structural role [44]. Summarizing, water interactions, both hydrophilic and hydrophobic, are thus key elements in determining its properties and functions in all material science, including biological materials where water is essentially in a confined state.

Recent NMR studies, made at the thermal denaturation of a hydrated protein (lysozyme), have clearly confirmed these suggestions, showing that the hydrophilic (the amide NH) and hydrophobic (methyl CH${}_{3}$ and methine CH) peptide groups evolve and exhibit different temperature behaviors. This clarifies the role of water and hydrogen bonding in the stabilization of protein configurations [45]. The data have also revealed the role of hydrophobic effects in this important protein intramolecular process and on water properties. These findings together with two NMR studies (the aforementioned confined water in carbon nanotubes [43] and a recent on water–methanol solutions [46]) inspired the present work that shows a detailed study of water properties in two different solutions: glycerol and methanol at many different water molecular fractions $X_{W}$ and in a very large temperature range (180–350 K), including the stable and the supercooled region, where polymorphism is relevant in determining the water properties.

In this frame, we considered the differences in the sizes of these amphiphiles, in their boiling points (glycerol 563 K and methanol 337.8 K), and in their overall thermal behavior: (i) the glycerol melting temperature is $T_{M}=291$ K so that, for the large part of the studied *T*, water is confined in a supercooled liquid, and viceversa, when methanol has $T_{M}=175.3$ K, the water is always in a solution as a liquid; (ii) the glycerol has a high molecular weight compared to that of water ($M_{WG}=92.112$ and $M_{WW}=18,015$ g/mol); and (iii) the molecular structure is C${}_{3}$H${}_{5}$(OH)${}_{3}$, made of three hydrophilic hydroxylic groups (OH), two external and one central, besides a central CH and the two external methylenes (CH${}_{2}$)-hydrophobic. The methanol ($M_{WM}=32.04$) is more simple (CH${}_{3}$OH), with a methyl group linked to a hydroxyl group).

This study essentially addresses the water behavior “confined” in these two liquids at different temperatures and concentrations, and the resulting effects of the hydrophobic–hydrophilic molecular competition and is developed by considering the available water transport functions (self-diffusion and relaxation times) as well as the measured specific heats of the three pure liquids. Specifically, we take advantage of the basic properties of the NMR technique to evaluate the behaviors of each molecular component of the studied system (at the same time) so that we can distinctly evaluate all of the hydrophilic and hydrophobic groups of the different molecules and their correlations.

In addition, from the self-diffusion, we evaluate both the configurational entropy $S_{conf}$ and the corresponding $C_{P,conf}$ in the frame of the Adam–Gibbs approach (developed for glass-forming supercooled liquids to clarify their cooperative relaxation processes [47]). Some important aspects of the molecular hydrophobic effect are clarified from the behaviors of the measured NMR spin–spin ($T_{2}$) and spin–lattice ($T_{1}$) relaxation times.

## 2. Data and Data Analysis

Figure 1 illustrates, in an Arrhenius plot (temperature range 400>T>165 K), the self-diffusion data, $D_{S}$, of pure water, glycerol, and methanol and their water solutions (many come from NMR and the other from the dielectric experiments and measured as relaxation times (DE)). Many data have been measured just for this work (specifically for the water–methanol solutions (Figure 1a) at $X_{W}=0.95,0.9,0.8,0.7,0.6,0.5,0.4$ for T<278 K; in the water–glycerol case (Figure 1b), the measured concentrations are $X_{W}=0.9,0.8,0.7,0.6,0.5,0.4$ for T<290 K). All other data come from the literature: for the methanol solutions, References [48,49,50,51,52,53,54,55,56] and, for those of glycerol, References [48,57,58,59,60,61]. For water, the reported data come from different experimental approaches: the bulk water data (reported as fully blue symbols) come from NMR experiments [62,63]; fused amorphous water (dark blue squares [22]) and MCM confined (actually measured NMR data are illustrated as dark blue squares and as open blue triangles [32]; and finally, the dielectric relaxation data [64] are illustrated as blue open circles).

We carried out NMR experiments using a Bruker AVANCE NMR spectrometer operating at a 700 MHz ${}^{1}$H resonance frequency and measured the $D_{S}$ using the pulsed gradient stimulated echo technique (${}^{1}$H-PGSTE), and the sample temperature stability was maintained in the range ±0.2 K. The inversion recovery pulse sequence was used to measure the spin–lattice relaxation $T_{1}$, varying the inter-pulse delay from microseconds to several seconds, and the spin–spin relaxation ($T_{2}$) was obtained from the Carr–Purcell–Meiboom–Gill procedure. The samples were prepared at the desired water molar fraction using pure glycerol and methanol (99.9%, from Fisher Scientific) and double distilled water.

From the illustrated $D_{S}$ data (Figure 2), two different behaviors are evident from the methanol and glycerol solution at different *T*. Whereas in the glycerol case (right panel), the behavior is regular and continuous (similar to that of an ideal mixture, a *T* change affects only the slope), for the methanol solutions (left panel) at the highest temperatures, in the range $1>X_{W}>0.5$, a decrease in the water content corresponds to a $D_{S}$ decrease with a minimum at $X_{W}≃0.5$. For methanol solutions, below a certain temperature in the water supercooled regime, the $D_{S}$ data change curvatures from concave to convex. These behaviors in the local dynamic data (such as the self-diffusion or the mean-square displacement) can be due to the hydrophobic effect, present in both solutions but with different results due to the different molecular sizes of the three substances, their thermodynamic status (stable liquid or supercooled), and the number of hydrophilic (OH) and hydrophobic (CH, CH${}_{2}$, and CH${}_{3}$) groups. Considering all this, their comparable molecular weights and the equivalent probability between a HB or a hydrophobic repulsion the water and methanol molecular dynamics are certainly very sensitive to the hydrophobic effect, when these substances are in solution. The opposite is true for glycerol–water solutions where three water molecules are in principle necessary to saturate the three OH groups of a single glycerol molecule with HBs. In any case, these effects should be opposed by the HB water network (and the LDL development and growth) by supercooling.

## 3. Results and Discussions

### 3.1. NMR Relaxation Time Data

This situation is well clarified from the temperature and concentration evolution of the measured $T_{2}$ and $T_{1}$ NMR relaxation times, reported in Figure 3 and Figure 4. The first one reports these times measured in the range 335>T>200 K for all three pure materials, where differences between the values of the different substances and different thermal evolutions are observable. It can be observed for both water and glycerol that, in some temperature ranges, the two times assume identical values: for water, this occurs at $T<T_{m}$, whereas for glycerol, this happens at higher temperatures. In the past, glycerol, being (similar to water) a sort of prototype of a glass-forming material, has been characterized using different experimental techniques, in particular NMR [65], or the combination of NMR and neutron scattering (inelastic and quasi-elastic) [66]. The $T_{1}$data proposed in the Figure 3 (as well as data measured in the glycerol supercooled regime [65,66]) have the same temperature behavior, characterized, as can be observed, by a well-defined minimum at about 280 K. It was also clearly noted that [65] (a) such a minimum is pressure- and temperature-dependent; (b) in liquid glycerol, the HB network is relatively insensitive to density changes but reflects only temperature effects; and (c) hydroxyl groups and the carbon backbone are strongly correlated. The situation suggests how the molecular dynamics of glycerol proceeds via a large-angle jump diffusion mechanism.

The main question proposed by the neutron NMR study was whether its molecular units C${}_{3}$H${}_{5}$ and O${}_{3}$H${}_{3}$, as essentially proposed in Figure 3, had the same time behavior or would behave differently when approaching the dynamic arrest. More specifically, a combination of ${}^{2}$H-NMR spin–lattice relaxation and quasi-elastic neutron scattering experiments (that measure the total mean square displacement ($\langle r^{2}\rangle$)—equivalent in a Brownian approximation to $D_{S}$ ($\langle r^{2}\rangle =2D_{S}t$) on deuterated glycerol (C${}_{3}$H${}_{5}$(OD)${}_{3}$ and C${}_{3}$D${}_{5}$(OH)${}_{3}$)) revealed that the corresponding measured $T_{1}$ have the same temperature behavior except beyond the glass transition (whereas the absolute values differ for a factor of ∼1.6) [66]. In particular, if normalized to the coupling constant of the bonds C-${}^{2}$H and O-${}^{2}$H, they assume identical values for $T>T_{g}$, indicating that the ${}^{2}$H spins of these bonds are subjected to the same motion. This study also stresses that the amplitudes and activation energies of C-bonded and O-bonded hydrogens are different, with the O–H motion being of a larger translational amplitude and a higher activation energy.

Upon considering the melting temperature of the three materials of interest, the studied data cover a wide region in which they are in the metastable state of supercooled liquids. However, the explored *T*-region does not include their glass transition temperatures, which are located at about 130 K for water, 103 K for methanol, and 190 K for glycerol. Certainly, the effects of the glass transition on the molecular mobility of these aqueous solutions ($T_{1}$, $T_{2}$, and $D_{S}$) are relevant and of interest (as well-known, in glass-forming materials at the $T_{g}$ extremes in the relaxation times and a dynamic crossover in $D_{S}$), but they are not among the objectives of this study, which is focused only on the HE.

Figure 3, instead, explains the different behaviors of the NMR relaxation times of the three materials. Those corresponding to water (reported only bulk water data) and methanol have values that decrease as *T* decreases and ranges, with comparable values, from 0.1–10 s. In the glycerol case, these times vary from $10^{-6}$ to 0.1 s, and the spin–lattice relaxation time has a minimum (∼0.8 ms at about 280 K, after than increases). For the water–glycerol solutions, NMR experiments ($T_{1}$ and $T_{2}$) made in the region 283–383 K fully confirmed the same linearity of the behavior as a function of the composition and temperature shown by the transport parameters [67,68]. It has also been shown that these solutions are essentially dominated by the HB interaction: the presence of water increases the overall glycerol mobility, and glycerol slows down the mobility of water [69]. Figure 3 also shows a different behavior between glycerol and methanol: while in the first case, the NMR relaxation times are the same for both the hydrophilic and hydrophobic groups, in the case of methanol, on the other hand, a different behavior is observed for the $T_{1}$ values of the groups OH and CH${}_{3}$. Their difference increases as the temperature decreases. This situation is of interest in the frame of the HE, and the fact that these two NMR times are close in magnitude to each other for water and methanol can lead to relative interferences when the two substances are mixed.

Being interested in the hydrophobic effect, we considered measuring the NMR relaxation times in the methanol–water solutions, also taking into account these comparable differences in the corresponding times of the pure liquids, considering the fact that some of them cross at a certain *T*. Therefore, we carried out their measurements in a mixture of the two substances even at the different molar fractions of water. The corresponding data for $X_{W}=0.95,0.9,0.8,0.6,0.5$, and 0.3 are reported in Figure 4, which illustrates the results obtained, with the top panel reporting $T_{1}$ and the bottom one reporting the spin–spin relaxation times. As observed, both for pure water and for the lower concentrations of methanol, the data are limited (compared to the other $X_{W}$) and stop at the lowest temperatures at the solidification point of the sample. The observable results in both figures are surprising and interesting as they propose some behaviors typical of bulk water as well as clarify some properties of the hydrophobic effect. Starting from $T_{1}$, top of Figure 4, the solution data in all contributions (OH${}_{W}$, OH${}_{M}$, and CH${}_{3}$) show a well-defined minimum at the temperature of the dynamic strong–fragile crossover of the water, i.e., $T_{L}≃225$ K, where the phase LDL and the HB networking [32,70] are dominant. Furthermore, the values and the behavior of the spin–lattice relaxation times of the two hydroxylic groups are identical, although smaller than those measured in bulk water, while those of the methyl groups are slightly larger. However, as reported in the bottom of Figure 4, the behaviors measured in the corresponding spin–spin relaxation times are more intriguing. What is immediately noticed is that the solution values are essentially lower than those of the pure components and that, at the temperatures of the stable liquid water, the values corresponding to the methyl group are higher (two orders of magnitude) than those of the two hydroxyl groups but, at the lowest *T* their behavior is identical with a maximum, just at $T_{L}≃225$, where all of the measured spin–lattice times have a minimum. More precisely, at high temperatures (while *T* decreases), the values corresponding at the two hydroxyls show a well-defined minimum located within the experimental error at ∼315 K. This temperature is a remarkable thermodynamic property for water: it is the place with the minimum isothermal compressibility ($\kappa_{T}\left(P,T\right)$) at all pressures and represents the point where all lines of the expansion coefficient ($\alpha_{P}\left(P,T\right)$) cross each other [71]. It has also been suggested that it is the locus of the onset of the HB tetrahedral structure [72,73].

With decreasing *T*, the $T_{2}$ dynamics appear to be weakly correlated up to about 265 K (indicated by the vertical blue line) for all of the studied concentrations; in contrast, when the methyl spin–spin times decrease, those corresponding to the hydroxyl groups (of the two substances) instead grow after showing a the minimum, just where, at T∼315 K, the onset of the HB network (and thus of the LDL phase) was suggested [71,72,73]. The subsequent decrease in temperature, on the other hand, involves these different groups: at about 265 K, regardless of their composition, there are two different singularities, with the hydrophobic methyl data having a minimum where the two hydrophilic groups have an inflection point. After that, upon approaching the deep supercooled regime, the methyl and hydroxyl times assume analogous thermal behavior, indicating a precise correlation between the three molecular groups of the solution. Furthermore, all of these data evolve at the same maximum located within the experimental error at the temperature of the Widom line where the water polymorphism is dominated by the LDL phase ($T_{L}≃227$ K).

This is a significant result; in other words, the reported data show that this latter temperature (265 K) represents a crossover for the hydrophobic effect: below it, the solution dynamics dominates in hydrophilicity (HB interaction), but above it, the hydrophobicity and its effects are certainly active and relevant. In other words, for T>265 K, the hydrophobic effect, or the dominance of the hydrophobic interaction with respect to the hydrophilic one, prevents the formation of structures between the solute and the solvent driven by the HB.

### 3.2. Configurational Effects

After obtaing these results, to better clarify the observed effects, we took into consideration the hypothesis of treating the self-diffusion data of both solutions (as well as of the three substances) with the Adam–Gibbs model by calculating the configurational contribution to their entropies and specific heats. In this context, the $C_{P}\left(T\right)$ values measured (Figure 5a) in the three substances both in the liquid phase (stable and supercooled) and in the solid phase are proposed as functions of temperature. Figure 5a, with the specific heats of glycerol [74], water [75,76,77,78], and methanol [79,80], shows (vertical dotted lines) their melting temperatures. Regarding water, we must stress that a very recent experiment considered the possible locus and intensity [81] at ambient pressure of the $C_{P}$ maximum at the Widom line, suggesting that it is located at about 229 K and has a value of 218 J/molK.

It can be therefore observed that, for glycerol and water, many values belong to the metastable state of supercooled liquid. The figure also shows that the specific heats of water and methanol cross at about 265 K. While below this temperature, those of water are greater than those of methanol, for higher values, it is methanol that has higher and increasing values. As theoretically proposed for water, we can assume [82] that the difference between the liquid and solid specific heat can give a good estimation of the configurational contribution, with such a difference for the three substances, $C_{P,conf}≃\Delta C_{P}=C_{P,liq}-C_{P,sol}$, evidenced in the Figure 5b. From the figure, we can observe that water and methanol $C_{P}\left(T\right)$ cross (due to the anomalous behavior of water) at about 265 K. In this frame, we must mention the considered methanol solid contribution that deals with the crystal I form of the two different crystalline states observed by Carlson and Westrum [80] and related to the classification of methanol as a plastic crystal on the basis of its small entropy of melting due to the HB effects.

The well-known Adam–Gibbs model (AG) is a molecular-kinetic theory, which explains the relaxation temperature dependence of glass-forming liquids. It was explained in terms of the T-variation in size of the cooperatively rearranging regions. In particular, it is shown that these sizes are determined by configuration restrictions, which can be expressed in terms of their configurational entropy. According to this, the transition probability $W\left(T\right)\;=\;Fexp\left(-z\Delta \mu /k_{B}T\right)$ of a cooperative region is evaluated as a function of its size *z* and Δμ (the potential energy-hindering cooperative rearrangements), where *F* is a frequency factor (negligibly T−dependent) and $k_{B}$ is the Boltzmann constant. By expressing the “critical size” $z^{*}$ of the cooperative region in terms of the molar configurational entropy $S_{conf}$, the transition probability can be expressed as $W\left(T\right)=Aexp\left(-C/TS_{conf}\right)$. With the system relaxation times related to the transition probability as τ(T)
$\propto W{\left(T\right)}^{-1}$, we have
(1)$$D_{S}\left(T\right)=D_{S0}\exp\left(-A/TS_{conf}\right)$$
wherer $D_{S0}$ and A=zΔμ can be assumed constant (at a given concentration). By using such an approach, the water configurational $C_{P,conf}$ was evaluated from the bulk water diffusion data (measured and simulated in the range 373–237 K) [82], obtaining $D_{S0}=1.07$ × $10^{-7}$ m${}^{2}$s${}^{-1}$ and A=31.75 kJ mol${}^{-1}$.

Very recently, the same analysis was made by considering the self-diffusion data of confined water using the same satisfactory values of $D_{S0}$ and *A* [83].

*A* and $S_{conf}$ can be thus evaluated by using Equation (Equation 1) in the bulk systems as far as all the water solutions are concerned, whereas a proper value of $D_{S0}$ was estimated from the $D_{S}\left(T\right)$ in the high *T* limit. Having in such a way obtained $S_{conf}$, the final step of the work was to calculate $C_{P,conf}$ as $C_{P,conf}=T{\left(\partial S_{conf}/\partial T\right)}_{P}$.Such a derivative was made after fitting the entropy data by means of a sixth-order polynomial in temperature. The top of Figure 6 reports the configurational entropy $S_{conf}$ for water, glycerol, and methanol (open symbols) as a function of *T*. For the water cases, we report also the contributions coming from confined (and fused) water measured well inside the supercooled region up to the amorphous phases, about 130 K (dark blue open circles and squares for confined and vapor deposited, respectively) [22,32,64]. In contrast, for glycerol and methanol, the data are reported in the temperature range of the corresponding diffusion data (see Figure 1). In the bottom of the Figure 6, the corresponding $C_{P,conf}$ data are illustrated; in particular, the data obtained according to the AG models from transport data are reported as open symbols whereas the measured specific heat are shown as full symbols. The obtained values of $D_{S0W}$ and $A_{W}$ are the same as those of the original simulation study, which gave for the first time evidence of a maximum in $C_{P}$ due to water polymorphism (maximum experimentally observed in confined water [84,85] and confirmed here by using the AG). For glycerol and methanol, $D_{S0G}=4.3$ × $10^{-7}$ and $D_{S0M}=6.7$ × $10^{-7}$ m${}^{2}$s${}^{-1}$, and $A_{G}=23.1$
$A_{M}=51.15$ kJ mol${}^{-1}$ are, respectively, obtained.

After these results, the solutions’ configurational entropies have also been evaluated from $D_{S}$ and the corresponding results are shown in Figure 7 (at the top are the water–methanol data, and that of water–glycerol are at the bottom). As observed in the methanol solutions, the $S_{conf}$ behavior at higher *T* (T>260 K) is not continuous: the value of pure water entropy is higher than that of the solutions in the $X_{W}$ range 0.9–0.4. This $S_{conf}$ behavior is, as proposed by the NMR findings on the spin–spin relaxation time, due to a local order, not related to HB networking but driven by the hydrophobic effect.

The entropy excess observed upon mixing water with hydrophobic species and the consequent non-ideal changes in other thermodynamic quantities were defined by Kauzmann [86] as challenging problems in the physics of aqueous solutions. The pioneering work of Frank and Evans [87] proposed the idea that hydrophobic entities enhance the water structure towards a more ordered one near the alcohol head groups. Lately, alcohol–water correlations have been studied using MD simulation [38,39,40,41,42,88] and experimentally [58,59,89,90,91] by confirming this local enhanced order. This indicates that, nowadays, there is a consensus that hydrophobic entities affect the water structure. The water pair correlation functions are sharpened compared to those of the pure liquid, with the consequent possibility of a hydration sphere around hydrophobic entities [89]. Finally, from the reported $S_{conf}$, by performing the corresponding derivatives for all of the studied pure materials and solutions, the configurational specific heats $C_{P,conf}$ have been evaluated (Figure 8). In particular, the configurational specific heat data evaluated for water, glycerol, and their solutions (at different molar fractions) (Figure 8a) and the corresponding data for water, methanol, and their solutions (Figure 8b) are reported as a function of the temperature. Figure also illustrates and compares the corresponding $\Delta C_{P}$ to that observed for $C_{P,conf}$ from pure glycerol, methanol, and water. In all three cases, the data obtained can be superimposed with the corresponding experimental values after multiplication, for a constant factor $C_{W},C_{Me},$ and $C_{Gly}$, but as can be seen, their *T* dependence is the same as that of $\Delta C_{P}$. An equal procedure is necessary for different water concentrations, but in this case, the used multiplicative factor is evaluated according to the molar fraction, as $C=X_{W}C_{W}+\left(1-X_{W}\right)C_{Me}$ (or $C_{Gly}$), and the obtained data are shown in Figure 8 (the error bar is the same as the symbol sizes). A significant result is that, for both solutions, a $C_{P,conf}$ maximum can be observed for samples in which we have $0.5<X_{W}<1$, indicating that the HB network and water polymorphism dominate the resulting structure for both solutions at the lowest *T*. Another observation comes from the data evolution of the methanol solution at high concentrations of methanol and the temperatures of the stable liquid water phase: the resulting $C_{P,conf}$ temperature evolution is analogous to that of methanol rather than to that of water, thus indicating, in accordance with NMR data, that methanol affects water caging. In particular, this shows how HE disfavors the HB network and, thus, the LDL phase as well as temperature increases.

This latter situation becomes clearer if we look at the total specific heat of the water–methanol system reported in Figure 9; the corresponding $C_{P}$ are obtained by adding the $X_{W}$ weighted values of the two solid phases to $C_{P,conf}$. What these data show finally clarifies that, in this high-temperature region, the behavior of the solutions is dominated by methanol. In fact, from the overall behavior of the reported data, a crossover temperature is evident (∼265 K), above which the methanol specific heat and those of the different solutions are higher than that of pure water, while in the opposite case, the specific heats for water and those of the solutions dominate on the corresponding methanol data.

## 4. Conclusions

Starting from the current research findings that water is also in the liquid phase dominated by a polymorphism generated by HB interactions and that this polymorphism, made from the LDL and HDL phases, as proposed by the LLT model [16], dominates the behavior of thermodynamic functions, also clarifying the origin of the anomalies that characterize the system, we evaluated the effects or interactions contrary to the formation of HB networking and, therefore, to the LDL phase. In this context, in order to adequately analyze the system in the supercooled regime, where the LDL dominates, we considered confined water. However, unlike water in nanotubes, we thought of solutions in which water remains liquid even at low temperatures, solutions with non-polar substances, which instead, possess both hydrophilic and hydrophobic groups, i.e., with a head that supports or promotes the HB interaction (and the corresponding networking) and a tail that is unfavorable to it and is, therefore, hydrophobic.

In such a way, it was therefore possible to study their contrasting effects on the structural and physicochemical properties of both the solvent and the solute on a molecular scale. In this context, we chose glycerol and methanol. The former together with water is a prototype of a supercooled glass-forming liquid with a comparative large molecular weight and a different internal structure made of three hydrophilic groups and as many hydrophobic tails. Methanol, instead, has a molecular weight comparable to that of water (and a similar mobility), with a hydrophilic and a hydrophobic head so that it can act alternatively either as an HB supporter or as its destroyer. With this approach, considering the water confined in the respective solutions (water–glycerol and water–methanol) and using an experimental technique, such as NMR, sensitive to both order and dynamics on a molecular scale, it was possible to clarify some aspects of the hydrophobic effect (which represents the central point of this study).

We evidenced the temperature evolutions of the spin–spin ($T_{2}$) and the spin–lattice ($T_{1}$) relaxation times as far as the self-diffusion coefficient ($D_{S}$) is concerned in the range 180–350 K. In the first case, we took advantage of the NMR property to follow the behaviors of each molecular component separately (at the same time), so that we distinctly measured all of the hydrophilic and hydrophobic groups of the different molecules (water, glycerol, methanol, and their solutions). The self-diffusion data obtained for these three transport functions, according to the ${}^{1}$H-PGSTE method used to deal with the motion of the entire molecule, were then compared with those reported in the literature.

While the relaxation time behaviors were analyzed directly, the data related to self-diffusion were studied according to the Adam–Gibbs model. In this latter case, the aim was to highlight the behavior of the configurational entropy and the corresponding specific heat contribution.

Operating in this way, the main result obtained from all of the studied quantities, with regard to the “confined” water in the two solutions, appears to be different in relation to the two interactions of interest (hydrophilic and hydrophobic). While in the first case (water–glycerol), the HB interaction appears dominant for all studied temperatures and concentrations, in the second case (water–methanol), the presence of two different temperature regions, dominated separately by hydrophilicity and hydrophobicity, is evident. The crossover temperature between these regions is located at ∼265 K. Such a situation also appears to be linked to the polymorphism of water and to the relative balance between the LDL and HDL phases. In fact, below this crossover temperature, where the LDL phase (and therefore the HB networking) develops and grows, the NMR relaxation times and configurational specific heat show extremes (maximum and minimum) just at the temperature of the supercooled liquid water dynamic crossover and of the Widom line (Figure 4, Figure 6, Figure 8 and Figure 9). Instead, above this temperature (265 K), where the HB becomes weak and less stable and therefore dominates the HDL phase, the hydrophobic effect determines the solution properties.

In conclusion, the main discovery of the present study is that, similar to HB, the hydrophobic effect is strongly T-dependent but affects the aqueous solution properties in opposite temperature regions. This latter fact is a situation of great importance regarding the properties of many macromolecular systems where water is confined. Such a competition between the basic interactions of the system can cause significant configurational evolutions by changing the thermodynamic variables.

## Figures and Tables

**Figure 1 ijms-22-07547-f001:**
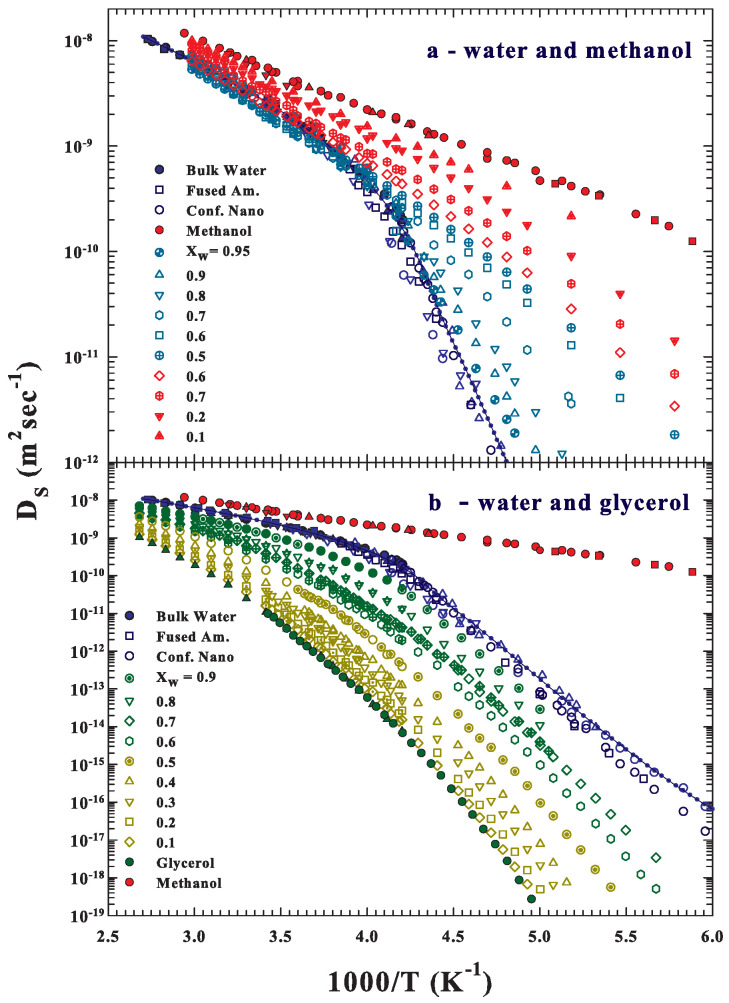
The self-diffusion, $D_{S}$, of pure water, glycerol, and methanol and their water solutions (many are measured by using NMR and the other from the dielectric experiments DE). The data are reported in one Arrhenius representation: log $D_{S}$ vs. 1000/T. Many data have been measured just for this work (specifically for the water–methanol solutions (**a**)) at $X_{W}=0.95,0.9,0.8,0.7,0.6,0.5,0.4$ for T<278 K. In the water–glycerol case (**b**), the measured concentrations are $X_{W}=0.9,0.8,0.7,0.6,0.5,0.4$ for T<290 K). All other data come from the literature: for the methanol solutions, References [48,49,50,51,52,53,54,55] and, for glycerol, References [48,57,58,59,60,61]. For water, the NMR data are proposed: (i) bulk water data (reported as fully blue symbols [62,63]); (ii) fused amorphous water (dark blue open squares [22]); (iii) MCM confined (actual data are proposed as dark blue squares and previous as blue triangles [32]; and (iv) finally, the dielectric relaxation data [64] (illustrated as blue open circles). Full and empty symbols represent pure materials and solutions, respectively.

**Figure 2 ijms-22-07547-f002:**
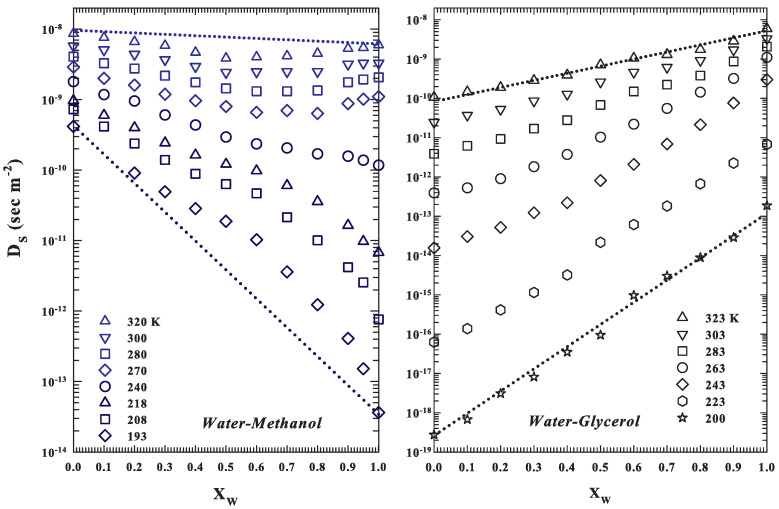
The two different behaviors in the D${}_{S}$ for the methanol and glycerol solution as a function of the water molar fraction at different *T*. The dotted lines indicate the behavior of a non-interacting ideal mixture. The situation, within the experimental error, is that of glycerol solutions (**right panel**), where a *T* change affects only the slope. Instead, for the methanol solutions (**left panel**), at the highest *T*, a decrease in the water content corresponds, within the range $1>X_{W}>0.5$, to a $D_{S}$ decrease with a minimum at $X_{W}≃0.5$. Again, for these latter solutions, below a certain temperature in the water supercooled regime, the $D_{S}$ data change curvatures from concave to convex.

**Figure 3 ijms-22-07547-f003:**
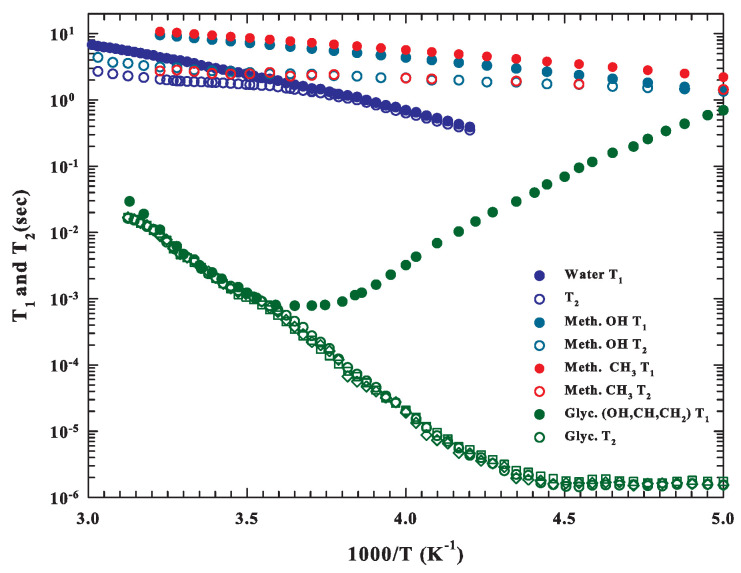
The Arrhenius representation of the measured spin–spin $T_{2}$ and spin–lattice $T_{1}$ relaxation times for water, methanol, and glycerol. For methanol and glycerol, the behavior of all of their different hydrophilic (OH) and hydrophobic groups (methine CH, methylenes CH${}_{2}$, and methyl CH${}_{3}$) are reported.

**Figure 4 ijms-22-07547-f004:**
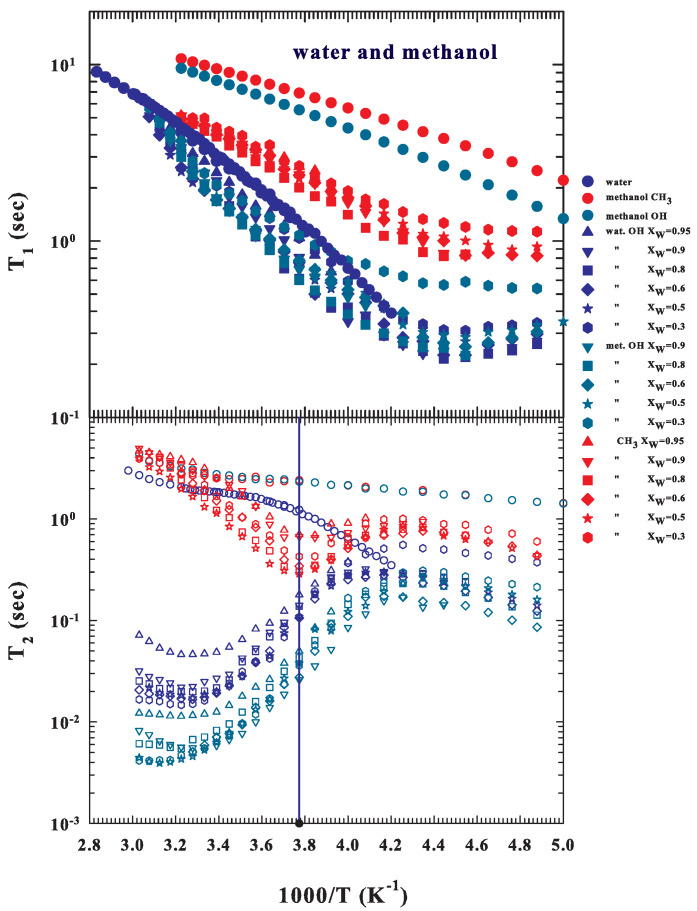
The Arrhenius representation of the NMR relaxation times $T_{1}$ (**top-filled symbols**) and $T_{2}$ (**bottom-empty symbols**) for water, methanol, and their solutions at the following water molar fractions: $X_{W}=0.95,0.9,0.8,0.6,0.5$ and 0.3 (the same symbol corresponds to each molar fraction). Here, we report the measured values of the hydrophilic (OH) groups of water (**blue symbols**) and methanol (**cyan**), and the methanol hydrophobic group (CH${}_{3}$, red), showing different behaviors. The vertical blue line on the bottom side indicates 265 K.

**Figure 5 ijms-22-07547-f005:**
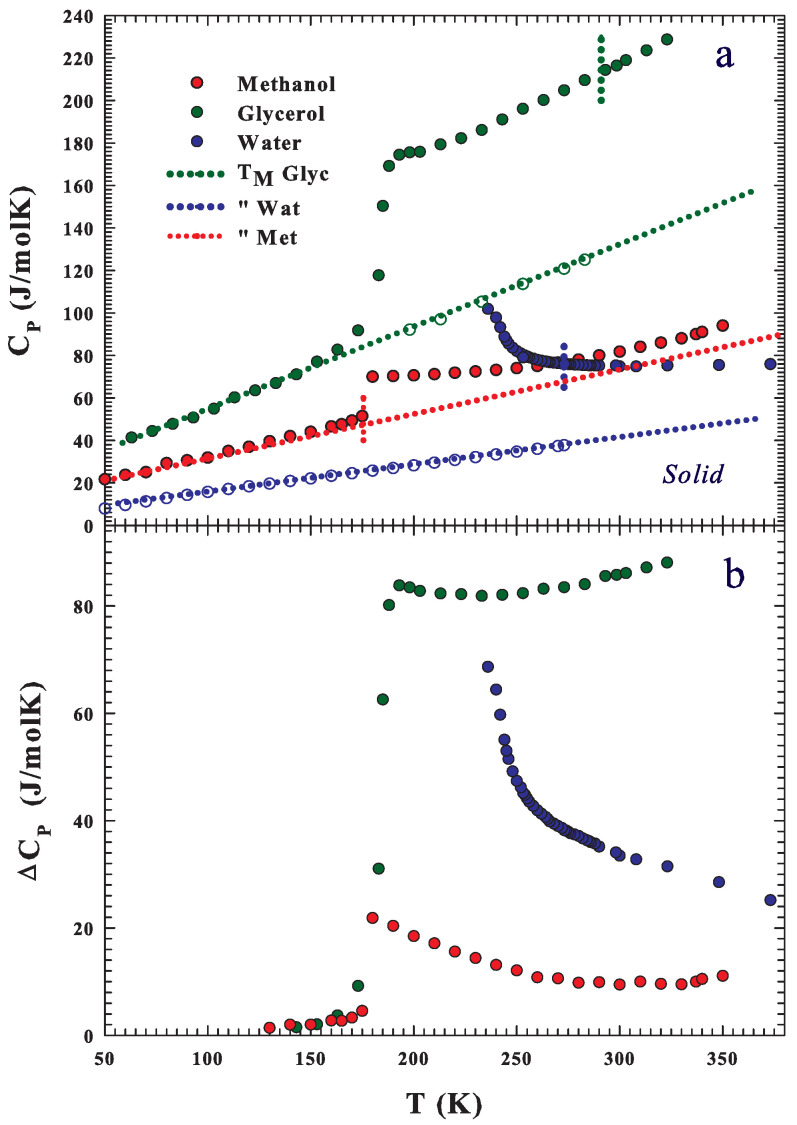
The water [75,76,78], methanol [79], and glycerol [74] $C_{P}\left(T\right)$ values measured (**a**) as a function of the temperature in their liquid (stable and supercooled) and solid phases (open circles and their interpolation as straight dotted lines). Their melting temperatures (vertical dotted lines) are also reported. As can be observed, the water and methanol specific heats cross at about 265 K. As theoretically proposed for water, we can assume [82] that the difference between the liquid and solid specific heat can give a good estimation of the configurational contribution; such a difference for the three substances, $C_{P,conf}≃\Delta C_{P}=C_{P,liq}-C_{P,sol}$, is proposed in (**b**).

**Figure 6 ijms-22-07547-f006:**
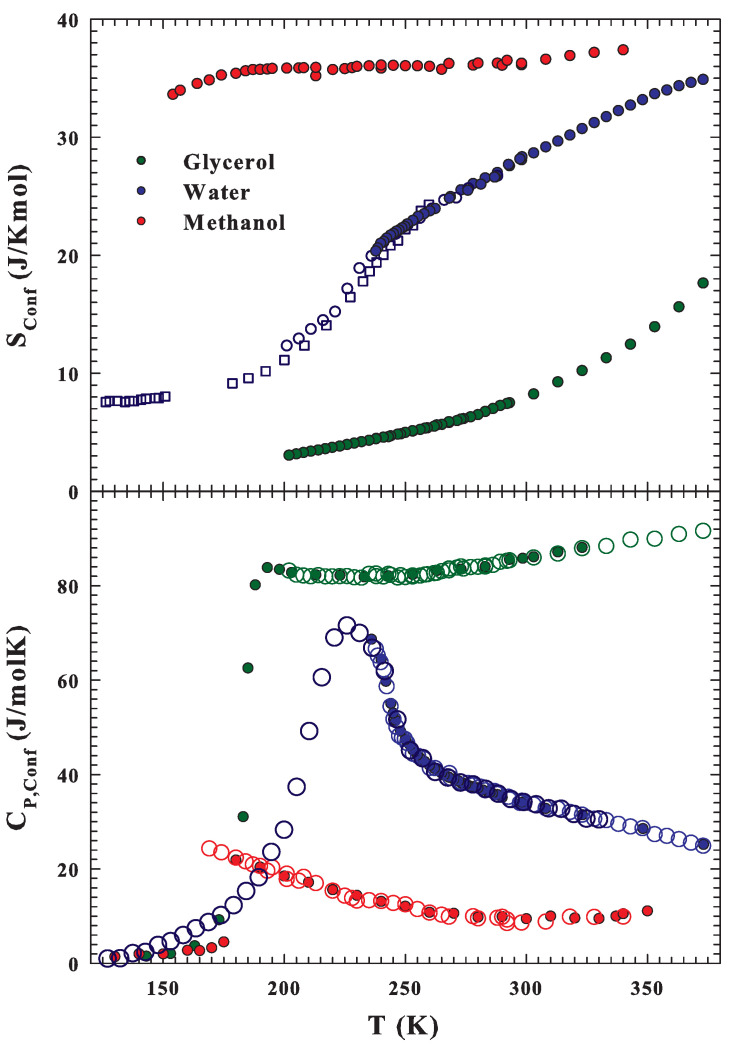
The water configurational entropies $S_{conf}$ (**top side**) and the corresponding specific heat contributions $C_{P,conf}$ (**bottom**) for water, glycerol, and methanol obtained in terms of the Adam–Gibbs model are reported as a function of the temperature. In the water cases, the contributions coming from confined (**circles**) and fused (**squares**) water, which allowed for measurement of the transport functions well inside the supercooled region up to the amorphous phases [22,32,64], are also reported as dark blue open symbols.

**Figure 7 ijms-22-07547-f007:**
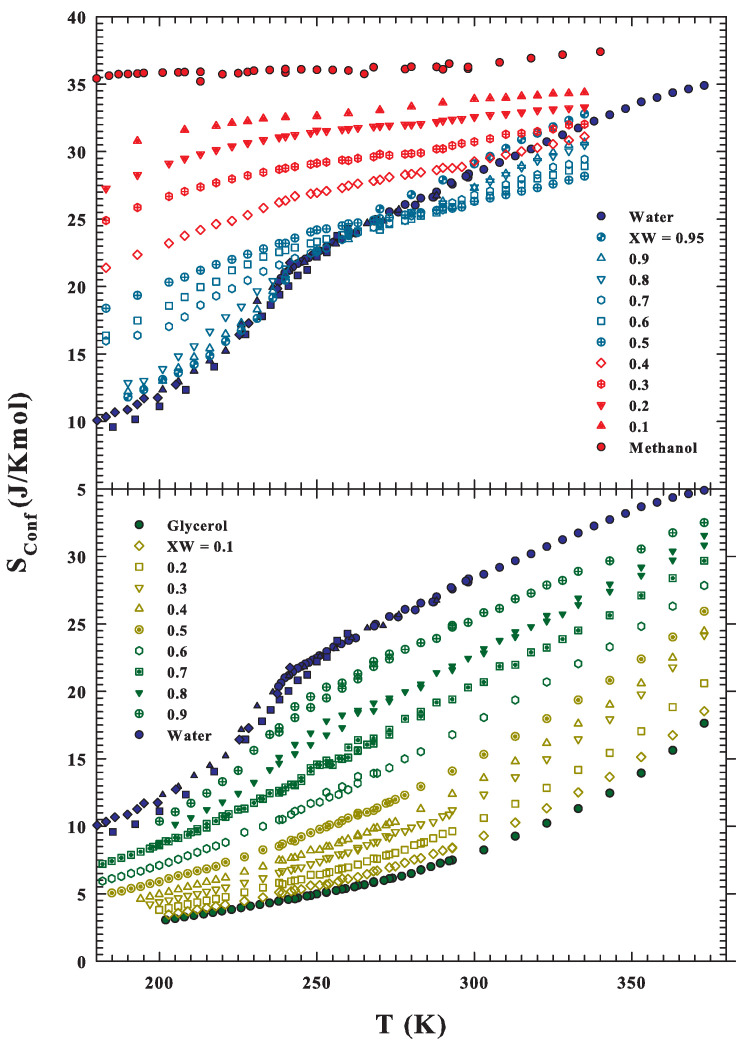
The solutions’ configurational entropies evaluated according with the Adam–Gibbs are proposed as a function of the temperature. At the top are the water-methanol data, and that of water–glycerol are at the bottom. As observed in the methanol solutions, their behavior at higher *T* (T>260 K) is not continuous with Xw: the pure water $S_{conf}$ is higher than that of solutions for the $X_{W}$ range 0.9–0.4. As explained, it is due to the hydrophobic effect as proposed by the NMR findings.

**Figure 8 ijms-22-07547-f008:**
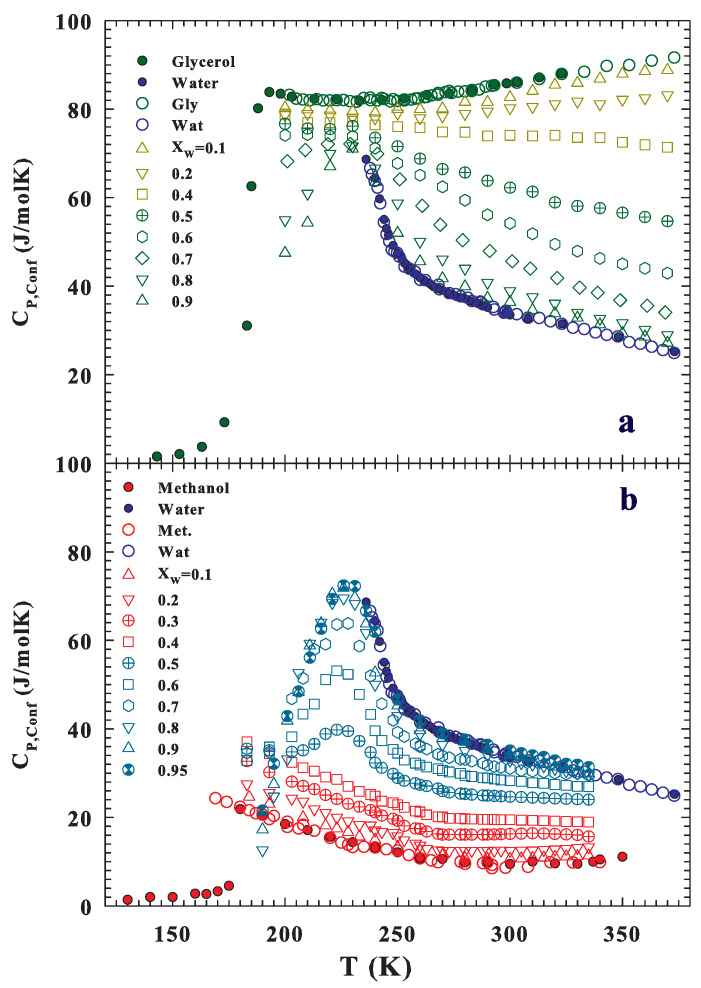
The configurational specific $C_{P,conf}$ (**a**) evaluated according to the Adam–Gibbs model for the solutions with water and glycerol at their different water molar fractions is illustrated. The bottom side (**b**), instead, shows the total specific heat of the water–methanol system, including their solution data and the same quantities evaluated for the water–methanol system. Both $\Delta C_{P}$ and $C_{P,conf}$ from pure glycerol, methanol, and water are also proposed.

**Figure 9 ijms-22-07547-f009:**
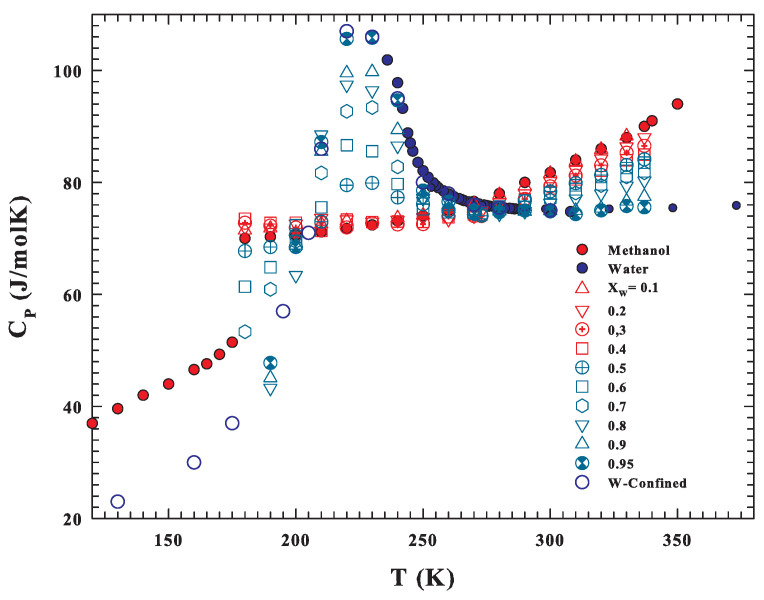
The figure show the total specific heat of the water–methanol system including their solution data.

## Data Availability

The data that support the findings of this study are available from the corresponding author upon reasonable request.

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
