# Peer review of "Hydrophilic and Hydrophobic Effects on the Structure and Themodynamic Properties of Confined Water: Water in Solutions"

_ijms, 2021, doi:10.3390/ijms22147547_

Round 1

Reviewer 1 Report

This manuscript includes an experimental investigation of hydrophilic and hydrophobic effects in aqueous solutions using NMR. Specifically the investigations include water-methanol and water-glycerol and solutions, covering the temperature range 180 K – 350 K. The results include self-diffusion coefficient and spin lattice T1 and spin-spin T2 relaxation times as a function of concentration and temperature.  Data analysis using the Adam-Gibbs model relates the results to the configurational entropy and isobaric specific heat capacity Cp. The extracted Cp is compared with that of confined water and the results are discussed in the connection to the hypothesized high and low density liquid water phases (HDL and LDL). The main conclusions are that: (a) In the case of water-glycerol solutions the hydrogen bond (HB) interactions is dominating the temperature/concentration dependent changes, whereas (b) in the case of water-methanol solutions the regime above/below 265K is mainly dominated by the hydrophobic/hydrophilic effects.

Overall, I found the experimental approach and analysis (especially the NMR analysis of different molecular groups, OH, CH3 etc., as well as the idea of studying hydrophobicity as a counter effect of LDL formation) very interesting. The results (particularly the difference in dynamics for hydrophobic and hydrophilic moieties and the crossover at ~265K) are intriguing as well and provide new insights into aqueous solution dynamics. My main concern is the language/formulations which could be much improved in order to improve readability, since several typos are present that can be improved with proof-reading (including the names of one of the authors in the online information). The results section could also use additional clarifications and structure as I found it difficult to follow sometimes for someone who’s not an expert in NMR relaxation time analysis, where the authors switch between different relaxation times (T1, T2) and solutions. Finally, regarding the figures, I found some of them a hard to interpret with the large number of symbols in a single plot  and I also noted that some symbol descriptions seem to be missing, whereas some figures could also benefit from being split up in two to a double panel.

Therefore, I believe that the current paper should be considered for publication in the International Journal of Molecular Sciences, after the following comments have been properly considered.

Major comments

  1. The authors conclude in the abstract that for water-methanol the region that the T>265K is dominated by hydrophilicity and T<265K by hydrophobicity. On the other hand in the conclusion section is stated that for T>265K the hydrophobic effect determined the solution properties. This should be clarified here, since the opposite conclusion is drawn.
  2. Page 7: “Between the two solutes, methanol, due to the methyl group, is certainly of high and effective hydrophilicity if compared to that of the glycerol.” Do the authors mean hydrophobicity here?
  3. Page 10. “These results evidence that the dynamics in a water-glycerol solution will be essentially dominated by the glycerol molecules with a linear behavior as that proposed in Figure 2.”. It would be useful here to explain in more detail the basis of this conclusion. The previous discussion was about the pure solutions of water and glycerol, not about the mixture.
  4. Page 11: “On decreasing T , the T2 dynamics are completely uncorrelated up to about 265 K (where the methyl spin-spin times have a minimum), after than on approaching the deep supercooled regime methyl and hydroxyls times assume identical thermal behavior “. This is a very interesting result, but what does it but what does it physically mean that the relaxation dynamics are different above 265K?
  5. Page 11: “the reported data show that this latter temperature (265 K) represent a crossover for the hydrophobic effect: below it, the solution dynamics is dominating by the hydrophilicity (HB interaction) but above it the hydrophobicity and its effects are certainly active and relevant. ”. Again here this is very exciting but what is the physical interpretation of hydrophobicity here? Hypothetically, how does the molecular arrangement change upon crossing over to the HB dominant region? This is also opposite to the conclusion in the abstract.
  6. Page 14: “evolution of the methanol solution at the temperatures of the stable liquid water phase: the resulting CP,conf temperature evolution is analogous to that of methanol rather than of water, thus indicating, in accordance with NMR data, some effect of methanol on the water caging.” Can the authors elaborate more on this interpretation and the consequences? How does this related to the HDL/LDL framework?
  7. Page 15: ”we took advantage of the NMR property to follow the behaviors of each molecular component separately (at the same time), so that we have measured distinctly all the hydrophilic and hydrophobic groups of the different molecules” I think this is a really interesting/advantageous experimental approach which could be more highlighted in the introduction.
  8. 8 is very difficult to read with the overlapping lines and symbols. It would be beneficial if the authors could split this into additional panels instead to increase readability.

Minor comments

  1. Page3: include reference in the sentence “For ambient pressure the diverging temperature was observed for κT at TS ∼”
  2. Page 4: include refernce in the sentence “it is estimated, by MD experiments, to be located near 200 K and at a pressure less than 200.064MPa.”
  3. Page 4: “Unfortunately, in spite of the very large number of accurate computational studies, with their positive and proper suggestions”. Reformulate the “positive and proper suggestions”
  4. Page 7: typo “In addition, form the self-diffusion we will” should be “from”
  5. Page 8” From the proposed Ds data”. What is proposed here? Do the authors mean measured?
  6. Page 9: typo: “This study also stres that”
  7. Page 10: “these times vary from 10−6 to 0.1”. Include units
  8. 1 caption: there are several issues here with the symbols. For example the fused amorphous water (is this open squares?). What is meant here by “actual data”?
  9. 2 caption: description of the dotted lines are missing here
  10. 5: description of the open circles is missing here
  11. 6: description of the open circles is missing here. Also why is the Sconf for glycerol shown in a shorter temperature range compared to Cp,conf since the latter is supposedly calculated from the former)?

Author Response

First of all a lot of thanks for you report. We have taken into account all your suggestions that we consider absolutely propositive and constructive. In particular, all changes made are reported in the new text in bold

Reviewer 2 Report

The manuscript by Mallamace et al. reports on NMR experiments of aqueous glycerol- and methanol solutions. The effect of the solute on spin-spin and spin-lattice relaxation times as well as the self-diffusion, is analyzed in terms of hydrophobic and hydrophilic interactions. Hydrogen bond interaction dominates the processes in the glycerol-water mixtures, instead the interaction for methanol-water solutions differs distinctly in the temperature region above and below 265 K.

The results show that hydrophobic interactions dominate at higher 
temperatures, which is consistent with the existence of predominantly high-density liquid water at elevated temperatures. The authors show a very comprehensive dataset of glycerol-water and methanol-water 
solutions, and evaluate the configurational entropy and specific heat capacity from the selfdiffusion data through the Adam-Gibbs model. In particular the finding of the two temperature regimes in the methanol-water system and its connection to the two states of liquid water is new and interesting.

Nevertheless, I do have some remarks and questions, that need to be addressed before publication.

Page 9 refers to glycerol as “a sort of prototype glass forming material”. What I miss in the discussion throughout the paper, is the effect of the glass transition temperature on T1, T2, and Ds on the different solutions. Tg of glycerol is located at 190 K, while Tg for methanol is at 103 K. The difference in the glass transition temperature will surely have an effect on the 
mobility of the different aqueous solutions. It is well known in literature that the minimum in T1 is characteristic for glass forming materials, as also seen in figure 3 for pure glycerol. So how about the minima observed for the methanol solutions in figure 4?

Page 11, part B: Please cross check the methanol data and also refer to other datasets. This is e.g., it was reported on two different crystalline states by Carlson & Westrum (JCP 54, 1971), further information, also on the glass transition temperature can be found in different articles by Angell et al.. This is, the authors should particularly state in figure 5, if they refer to a glassy 
or crystalline state, when they talk about the solid states. Looking at delta Cp might be misleading otherwise, as it is the glass transition in glycerol that is showing the huge increase in Cp, while melting of methanol takes place at a similar temperature with smaller delta Cp.

Page 11, last line: eventually add “…of water and methanol cross due to the anomalous behavior of water at about 265 K”.

Conclusion on pager 15: I find the two temperature regimes present in the water-methanol solution quite convincing as probe for the hydrophobic effect. However, I would like to ask the authors to describe more specific the difference they see for the OH and CH3 groups. This aspect is not clear to me, can you refer directly to the figure and comment on this? E.g., in 
figure 3 I can´t see any difference between T1 or T2 of OH, CH3 and CH2, respectively.

Minor points:

Page 3: I am wondering why the introduction refers to ice XII, even though meanwhile XIX states of ice have been detected.
In the same paragraph there is some hiccup in the sentence about pressure-amorphization, as it was LDA in the form of vapor deposited ice that was known since 1935, while HDA was discovered by Mishima 1984. He also referred to the HAD-LDA transition as first-order-like.

Page 10; 8th-last line: Figure …. Number is missing.

Page 13, second paragraph, first line: eventually add “…have been evaluated from Ds …”

Page 13, middle: please provide a reference for the statement of pair correlation functions.

Figure 1: - Shouldn’t it be (m^2 / s)
- Filled green triangle are not explained
- In the figure caption dark blue square are describe both as vapor deposited ice as well as MCM confined water
- Similar to figure 2 it would help to have “Water-Methanol” and “WaterGlycerol“ directly written in the figure.

Figure 2: The minimum in the water-methanol HT-data was already observed by Derlacki et al. (JPC 89, 5318, 1985).

Figure 4: Not all symbols are explained.

Figure 5: As reference for the supercooled water data, the authors should include “Angell et al., JPC 86, 998, 1982. It might also be interesting to discuss recent findings by Pathak et al.(PNAS 118, 2021), that showed Cp for water at even lower temperatures could potentially rise above the value of glycerol.

Figure 6: Make clear which data are from confined water and which from vapor deposited ice.

Author Response

The manuscript by Mallamace et al. reports on NMR experiments of aqueous glycerol- and methanol solutions. The effect of the solute on spin-spin and spin-lattice relaxation times as well as the self-diffusion, is analyzed in terms of hydrophobic and hydrophilic interactions. Hydrogen bond interaction dominates the processes in the glycerol-water mixtures, instead the interaction for methanol-water solutions differs distinctly in the temperature region above and below 265 K.

The results show that hydrophobic interactions dominate at higher temperatures, which is consistent with the existence of predominantly high-density liquid water at elevated temperatures. The authors show a very comprehensive dataset of glycerol-water and methanol-water solutions, and evaluate the configurational entropy and specific heat capacity from the self-diffusion data through the Adam-Gibbs model. In particular the finding of the two temperature regimes in the methanol-water system and its connection to the two states of liquid water is new and interesting.

Nevertheless, I do have some remarks and questions, that need to be addressed before publication.

We report or answers by referring to the pdf file of the manuscript in the IJMS format (version June 26)

Page 9 refers to glycerol as “a sort of prototype glass forming material”. What I miss in the discussion throughout the paper, is the effect of the glass transition temperature on T1, T2, and Ds on the different solutions. Tg of glycerol is located at 190 K, while Tg for methanol is at 103 K. The difference in the glass transition temperature will surely have an effect on the mobility of the different aqueous solutions. It is well known in literature that the minimum in T1 is characteristic for glass forming materials, as also seen in figure 3 for pure glycerol. So how about the minima observed for the methanol solutions in figure 4?

Our answer -from line 234 to line 240 of the pdf file - :

On considering the melting temperature of the three materials of interest, the studied data cover a wide region in which they are in the metastable state of supercooled liquids. However, the explored T-region does not include their glass transition temperatures which are located: at about 130 K for water, 103 K for methanol and 190 K for glycerol. Certainly the effects of the glass transition on the molecular mobility of these aqueous solutions (T1, T2 and DS) are relevant and of interest (as it is well known glass forming materials show at the Tg extremes in the relaxation times and a dynamic crossover in DS), but they are not among the objectives of this study focused only on the HE.

Page 11, part B: Please cross check the methanol data and also refer to other datasets. This is e.g., it was reported on two different crystalline states by Carlson & Westrum (JCP 54, 1971), further information, also on the glass transition temperature can be found in different articles by Angell et al. This is, the authors should particularly state in figure 5, if they refer to a glassy or crystalline state, when they talk about the solid states. Looking at delta Cp might be misleading otherwise, as it is the glass transition in glycerol that is showing the huge increase in Cp, while melting of methanol takes place at a similar temperature with smaller delta Cp.

Page 11, last line: eventually add “…of water and methanol cross due to the anomalous behavior of water at about 265 K”.

Our answer for both the questions -from line 317 to line 321 of the pdf file - :

From the figure can be observed that water and methanol CP(T) cross (due to the anomalous behavior of water) at about 265 K. In this frame, we must mention the considered methanol solid contribution that deals with the crystal I form, of the two different crystalline states observed by Carlson and Westrum [79], and related to the classification of methanol as a plastic crystal on the basis of its small entropy of melting due to the HB effects.

Conclusion on pager 15: I find the two temperature regimes present in the water-methanol solution quite convincing as probe for the hydrophobic effect. However, I would like to ask the authors to describe more specific the difference they see for the OH and CH3 groups. This aspect is not clear to me, can you refer directly to the figure and comment on this? E.g., in figure 3 I can´t see any difference between T1 or T2 of OH, CH3 and CH2, respectively.

Our answer -from line 283 to line 294 of the pdf file - :

On decreasing T, the T2 dynamics, for all the studied concentrations, appear to be weakly correlated up to about 265 K (indicated by the vertical blue line); whereas the methyl spin-spin times decrease, those corresponding to the hydroxyl groups (of the two substances) instead grow 285 after showing a the minimum just where, T ~ 315 K, the onset of the HB network (and thus of the LDL phase) was suggested [70–72]. The subsequent decrease in temperature, on the other hand, involves among these different groups: at about 265 K, regardless of their composition, two different singularities: the hydrophobic methyl data have a minimum just where the two hydrophilic groups have an inflection point. After than on approaching the deep supercooled regime the methyl and hydroxyls times assume analogous thermal behavior, indicating a precise correlation between the three molecular groups of the solution. Furthermore, all these data evolve at the same maximum located, within the experimental error, at the temperature of Widom line where the water polymorphism is dominated by the LDL phase (TL @ 227 K).

Minor points:

Page 3: I am wondering why the introduction refers to ice XII, even though meanwhile XIX states of ice have been detected.
In the same paragraph there is some hiccup in the sentence about pressure-amorphization, as it was LDA in the form of vapor deposited ice that was known since 1935, while HDA was discovered by Mishima 1984. He also referred to the HAD-LDA transition as first-order-like.

Our answer: We presume that the mentioned hiccup has been resolved by rewriting the corresponding period from line 40 to line 45.

Page 10; 8th-last line: Figure …. Number is missing.

Page 13, second paragraph, first line: eventually add “…have been evaluated from Ds …”

Page 13, middle: please provide a reference for the statement of pair correlation functions.

Our answer: all these errors have been corrected and the reference has been entered as 87.

Figure 1: - Shouldn’t it be (m^2 / s)
- Filled green triangle are not explained
- In the figure caption dark blue square are describe both as vapor deposited ice as well as MCM confined water
- Similar to figure 2 it would help to have “Water-Methanol” and “Water-Glycerol“ directly written in the figure.

Our answer: We believe we have done everything following the referee's suggestions.

Figure 2: The minimum in the water-methanol HT-data was already observed by Derlacki et al. (JPC 89, 5318, 1985).

Figure 4: Not all symbols are explained.

Our answer: The Derlacki study is reported as reference 55 and the symbols have been explained.

Figure 5: As reference for the supercooled water data, the authors should include “Angell et al., JPC 86, 998, 1982. It might also be interesting to discuss recent findings by Pathak et al. (PNAS 118, 2021), that showed Cp for water at even lower temperatures could potentially rise above the value of glycerol.

Our answer: Austen Angell's work is reported as a reference 76, while the one of Pathak is ref. 80.

Round 2

Reviewer 2 Report

Three of my initial questions I could not find answered and would kindly ask for clarification. 

  1. How about the minimum in T1. Such a minimum was also shown in the cited ref 65 for glycerol at temperatures above Tg, and I would like to see this discussed in the manuscript.
  2. Which differences do you see for the OH and CH3 groups ?
  3. Figure 6: Please make clear which data are from confined water and which from vapor deposited ice.

Author Response

We made the minor revisions proposed by referee 2. In particular, we have discussed the minimum in the T1 of glycerol
highlighting what is proposed in reference 64 (Wolfe, M.; Jonas, J. Chem. Phys. 71, 3252 (1979) where this spin-lattice
relaxation has been studied as a function of both temperature and pressure.